# Experimental Assessment of Time-Limited Operation and Rectification of a Bridge Crane

**DOI:** 10.3390/ma13122708

**Published:** 2020-06-14

**Authors:** Peter Frankovský, Ingrid Delyová, Peter Sivák, Piotr Kurylo, Elena Pivarčiová, Vojtech Neumann

**Affiliations:** 1Faculty of Mechanical Engineering, Technical University of Košice, Letná 9, 042 00 Košice, Slovakia; peter.frankovsky@tuke.sk (P.F.); ingrid.delyova@tuke.sk (I.D.); peter.sivak@tuke.sk (P.S.); Vojtech.Neumann@grob.de (V.N.); 2Faculty of Mechanical Engeenering, University of Zielona Góra, 65-516 Zielona Góra, Poland; P.Kurylo@ibem.uz.zgora.pl; 3Faculty of Technology, Technical University in Zvolen, Zvolen, Masarykova 24, 960 01 Zvolen, Slovakia

**Keywords:** lifetime of structure, crane, experimental analysis

## Abstract

This paper describes a problem related to a casting bridge crane with a combined load of 200/50/12.5 t and a span of 18.6 m, working in a heavy metallurgical operation. Due to the specific stress of the structure after its long-term operation, longitudinal fillet welds between the upper flange and the web of the main box beam on the rail side of the 200 t trolley were irreparably damaged. As a result, the cross-section of the main beam had opened, thereby substantially reducing its strength and stiffness. This resulted in a disproportionate increase and undesirable redistribution of stresses in the beam and, at the same time, an increase in the probability of acute fatigue or the loss of stability of the elastic beam shape. Therefore, the rectification of the damaged load-bearing structure was carried out by specific structural modifications. Critical load-bearing elements were subjected to complicated strength and fatigue life analyses before and after rectification. These analyses were supported by experimental measurements. The applied modifications resulted in a partial strengthening of the lifting device with the possibility of its further operation, but only in a limited mode, with a limited period of operation with a time limit of 2 years and a reduced total load capacity of 150 t. The applied methods are also applicable for the fatigue analysis of load-bearing elements and equipment for bridge, gantry and tower cranes, crane tracks, road and railway bridges and support structures under machinery and other devices with a dominant transverse and rotating effect.

## 1. Introduction

A serious and complex technical problem in steel structures is the assessment of their technical capability. Assessing the capability of load-bearing elements and structures for long-term operation and predicting their total or residual service life is a serious and very complex technical problem. 

In addition to the results of service life analysis, the final assessment of the state of the structure cannot be done without taking into account the results of another series of examinations, assessing mainly the loss of material due to corrosion and the change in mechanical properties of the structure, for instance with regard to the effect of temperature. In some cases, it is also necessary to consider the effect of residual stresses and check the strength, permissible deformations and stability of the structure and assess these results in the light of current technical regulations.

Based on the final assessment of the state of the structure and the fulfilment of all required or monitored parameters, an alternative to further exploitation in its original or barely modified form can be agreed upon, mostly with a guaranteed lifetime, expressed as a number of working cycles, hours, days or years, while observing predetermined conditions for the operation and maintenance of the whole equipment. 

If the equipment of the load-bearing structure does not prove to have a sufficient life span, or if this life span even proves to have been exceeded, decommissioning is proposed. In justified cases, e.g., after the occurrence of the fatigue damage limit state, a form of structural or technological modification or repair for the components in question or their complete overhaul is suggested. The entire structure is then additionally subjected to strength and durability analyses.

The durability and reliability of a crane’s constructional parts are the most relevant factors that are monitored during the operation of a bridge crane. Kopnov [1] investigated the fatigue durability of a gantry crane used in the forest industry. Qi et al. [2] evaluated the safety and analysed the fatigue durability of crane construction. Rettenmeier et al. [3] addressed the assessment of crack propagation in crane runway girders being travelled over by wheel loads, and in [4] addressed a fatigue evaluation procedure to assess the influence of multiaxial stress states caused by both traveling crane wheel loads and welding residual stresses on the lifetime of crane runway girders. Euler et al. [5] dealt with the fatigue evaluation of crane rail welds using local concepts. Černecký et al [6] studied beams and their models using holographic interferometry.

All types of lifting equipment, including overhead cranes, face an increased risk of fatigue damage. Typically, this equipment includes machines operating in metallurgical plants in a heavy-duty mode, with excessive wheel pressure and a large static and dynamic load effect. Also specific to such types of equipment is the unevenness of the crane track, crane cross movements, etc. Particularly sensitive to fatigue damage are, e.g., the parts of load-bearing structures near the crane trolley. In the case of an acute risk of fatigue damage limit state, it is possible to propose a method of rectification for such structural elements. After rectification, the whole structure can again be subjected to strength and durability analysis.

### Factors Determining Fatigue Damage

Fatigue damage can affect the structural elements and structural nodes of load-bearing structures due to unsuitable structural design during manufacture or unsuitable additional structural modifications, e.g., overhaul (project design factors), failure to observe technological discipline during the manufacture or repair of the equipment (manufacturing technology factors), the intentional or unintentional use of the equipment in a manner inconsistent with its intended use, including the unavoidable effects of natural factors (streaming wind and water, tectonic movement of the Earth) or incorrect so-called non-prescribed maintenance (operational load factors) [7,8,9,10,11].

Due to these factors, the real nature of the stress is then far more unfavourable than expected in the design of the structure, and the direction, gradient, magnitude, or redistribution of stresses may change. There is a higher risk of fatigue damage (failure) in such affected areas, elements and construction nodes [12,13,14]. 

In terms of ultimate fatigue failure, the following points and structural elements on the structure can be identified as critical:elements with unforeseen inconsistencies in their cross-sectional parts or connecting and joining elements with consequent undesired stress redistribution;elements with extra-jaw or off-axis loads associated with the formation of additional stresses;components or complete equipment for which the actual nature of the load or use is different from that prescribed;locations with sudden (step-like) cross-sectional changes;sites with a small transition radius;places with unsuitable roughness and accuracy deviation on machined surfaces [15,16];undersized elements with asymmetrical construction;attachment points of secondary elements to main elements at stress concentration points;combinations of two or more elements, arranged in parallel to each other, with extremely different levels of stiffness (combinations of a rigid and a flexible path);fatigue elements that are undersized but not functionally backed up;elements operating at maximum load during each duty cycle;elements or locations within them with high residual stresses arising from their manufacture or introduced after their mutual assembly [17];elements with extremely long overlaps and a large ratio of length to diameter [18];sheets with an abnormally small bending radius;reinforced elements, especially sheets, without relief cut-outs;connection points with an inappropriate combination of connecting elements, e.g., a combination of rivets and screws;eccentrically loaded mounting eyes, or eyes subjected to a high axial load in combination with a high compression load;threaded bolts (instead of rolled bolts) and preloaded bolts without the introduction of permanent pre-load;places of poor quality, or poor-quality tensile weld;elements of small thickness when joined by a butt weld;elements not protected against surface or contact corrosion;elements with a superheated surface due to high-speed grinding;incorrectly plated elements which subsequently undergo possible hydrogen brittleness or stress corrosion and the associated process of corrosion fatigue;non-interconnected elements between which there is wear-related play;elements which have undergone major deformations due to prolonged high temperatures;elements neglected with respect to regular or extraordinary inspection and maintenance.

Fatigue cracks can come from weld ends, places with sudden changes in their cross-sections, cut-off corners and other stress concentrators, sometimes also in conjunction with unsuitable quality and properties regarding the material used (steel). In particular, structural inhomogeneity, the use of low-impact steel, especially at low temperatures, low plasticity and/or low viscosity properties can play a negative role. The same negative effects cause other forms of steel degradation as a result of material ageing [19,20,21,22,23].

Achieving ultimate fatigue damage or structural element failure depends on the dynamics of the damage accumulation, which is a function of the sub-structural and structural state of the material, technological and construction characteristics of the product, its conditions of use and, above all, the time and magnitude of the affecting factors which give rise to a specific fatigue state [24,25]. The level of damage to the material and structural elements is characterised either by the level of internal energy, in particular at the concentration points, or by the volume fraction of the areas of material cohesion failure due to external or internal fatigue limit factors [26,27,28,29,30]. 

## 2. Main Starting Points for Determination of Structural Life

Ensuring the required service life and reliability of a structure during cyclic loading requires a number of activities of theoretical and experimental nature. When analysing fatigue life, in the simplest cases only one-stage deterministic load is considered. Fatigue life is most influenced by the presence of notches that cause stress concentration. Structural elements are often dimensioned assuming that the oscillating load takes a harmonic course. The additional introduction of operational factors is insufficient for taking into account the actual occurrence of load peaks and accidental increases in force. All calculation methods that take into account variable loads only indirectly are included in the notion of quasi-static procedures. If fatigue test results are not available, the service life curve is determined empirically. Such a quasi-static approach to the calculation of fatigue strength in the area of high-cycle fatigue represents a standard calculation rule according to which the fatigue resistance calculation is performed for load-bearing elements, fasteners and welds. Design fatigue strengths are determined by the operating group and depend on the load spectrum and the number of load cycles. They are given for different steels, stress types and notch groups and low and high stress ratios. At multiaxial stress, the strength condition must be met
(1)$${\left(\frac{\sigma_{x}}{R_{xfat}}\right)}^{2}+{\left(\frac{\sigma_{y}}{R_{yfat}}\right)}^{2}-{\left(\frac{\sigma_{x}\cdot \sigma_{y}}{\left|R_{xfat}\right|\cdot \left|R_{yfat}\right|}\right)}^{2}+{\left(\frac{\tau }{\tau_{fat}}\right)}^{2}\leq 1.1,$$
where $\sigma_{x},\;\sigma_{y}$,τ are the respective normal and shear stresses and $R_{xfat},\;R_{yfat},\tau_{fat}$ are design stresses corresponding to the design strength [7,14].

## 3. Analysed Object and Problem Solved

The analysed object was a casting bridge crane with a combined load of 200/50/12.5 t and a span of 18.6 m. Its supporting steel structure is shown in Figure 1. The coded specification 200/50/12.5 t corresponds to a bridge crane where the main part of the supporting structure consists of two outer girders (A, B, Figure 1), a crane crab with load capacity of 200 t and two inner girders including a crane crab with a combined load capacity of 50 t and 12.5 t. The crane had been operating in the mixing hall of the steelworks of a heavy metallurgical operation for 29 years. After this time, it was subjected to analysis and assessment of the condition of the load-bearing structure and the accumulation of fatigue damage at the weld flange and the web under the crane rail on the main beams. The main sources of information for the relevant analyses were, among other things, the results of experimental measurements based on an electrical resistance strain gauge. Normally, 200,000 to 600,000 cycles are expected for a given crane type. With 90% crane load, this accounts for 326,250 duty cycles in 29 years of operation. These data were calculated based on information provided by the operator. The limit number of cycles for the given crane based on the experimentally determined course of operational accidental stress and using fatigue damage accumulation hypotheses was set at 285,714 duty cycles. The fatigue life of the main beams was therefore both theoretically and practically exhausted. This was also confirmed by cracks occurring in longitudinal welds between the upper flange and the web of the main box beam on the side of the 200 t trolley. The positions and lengths of the fatigue cracks in the welds along the main beam are shown in Figure 2. As a result, the cross-section of the main beam (Figure 3a) had opened, thereby substantially reducing its strength and stiffness. This had resulted in a disproportionate increase and redistribution of stresses in the beam and, at the same time, an increase in the probability of acute fatigue or the loss of stability of the elastic beam shape.

## 4. Main Beam Rectification for Restricted Operation Mode

For further operation of the crane, with a reduced load capacity of 150 tons, the main beam of the crane was required, meaning that the main lift beam needed to be closed. Therefore, it was proposed that the beam should be rectified by structural modification according to Figure 3b [14].

The cross-section of the main beam was closed by welding a bent sheet along the entire length of the main beam between its outer vertical reinforcing ribs, Figure 3b. By adjusting the cross-section, the geometrical characteristics of the main beam were changed. The geometrical characteristics of the main lift beam before and after the main beam cross-section adjustment are given in Table 1.

The designed profile was computationally checked and the fatigue and stability check was also performed. The main beam was made of EURO S235JRG2 (ISO Fe360B). The specified design strength of this material is $R_{d}=210\;\;\mathrm{MPa}\;\;$ for thicknesses of up to 25 mm and $R_{d}=200\;\mathrm{MPa}\;\;$ for thicknesses above 25 mm. The external loading of the structure is caused by

its own weight;the actual load on the hook;jamming;the effects of inertia during run up and braking.

All these loads were considered in the calculation using partial factors listed in Table 2.

For the strength control of the modified cross-section of the main beam, internal force values calculated using a combination of the beam’s own weight, constant load and inertia loads were used. The analytically gained internal forces shear force in the z-direction, $T_{Z}$; bending moment to the y-axis, $M_{y}$;bending moment to the z-axis, $M_{z}$
were corrected to the values listed in Table 3 after the relevant factors (Table 2) were taken into account. Table 3 also lists the calculated normal, shear and reduced stresses σ, τ, $\sigma_{red}$ in cross-sections in the centre of the span of the main beam and at the point of its connection with the crossbeam in non-crossbeam mode and with the crossbeam on its track. Reduced stresses were calculated by adjusting relation (1) according to [7].
(2)$$\sigma_{red}=\sqrt{\sigma^{2}+3\tau^{2}}.$$

The strength condition at reduced stress values was
(3)$$\sigma_{red}\leq \;1.1\;R_{d}=231\;\mathrm{MPa}.$$

The calculation of the reduced stresses in the centre of the main beam and at the extreme end of the beam revealed that the main beam met the basic load combinations and that the reduced stresses did not exceed the permissible stresses. The modified cross-section of the main beam was not satisfactory after the fatigue assessment and it was therefore concluded that the structure was capable of limited operation, with a reduced load capacity of 150 t and a time limit of two years.

## 5. Assessment of Structural State Based on Experimental Measurements

During the study, strain gauge checks were carried out on the main beam with a modified cross-section. A strain gauge was used to determine strain at selected points on the main beams. A diagram showing the positioning of strain gauges S1 to S5 is shown in Figure 1. 

A measurement chain was formed by the dynamic tensiometric apparatus as shown in Figure 4.

Experimental measurements were repeatedly performed at the load weights 0 kg, 123,200 kg and 211,000 kg, and in three modes:mode 1—movement of the trolley from the centre of the main beam to the extreme positions of the beam and return to the centre;mode 2—trolley approximately in the centre of the main beam, loading of the lift, travel of the trolley to the front and rear cross members, return of the trolley to the starting position, lowering of the load;mode 3—trolley approximately in the centre of the main beam, loading of the lift, travel to the bridge, return to the starting position, travel to the front and rear cross members, return to the starting position, lowering of the load.

The maximum values of normal stress oscillation were found on the right main beam at strain gauge S3, Figure 1. Figure 5 represents the normal stress curve obtained from the measured values of strain at location of strain gauge S3 on the main beam before correcting the cross-section at maximum load in mode 3. 

At measuring point S3, i.e., on the upper flange, a uniaxial stress state was assumed with the direction of the principal stress identical to that of the longitudinal axis of the beam. The values of the normal stresses within the range of elastic deformations were calculated from the measured strain ε for the modulus of elasticity E and Poisson’s steel number μ, i.e., for $E=2\times 10^{5}\;\mathrm{MPa}$, μ=0.3. 

The normal stress was calculated based on the measured values (strain $\varepsilon_{x},\;\varepsilon_{y}$) according to the relation
(4)$$\sigma_{x}=\frac{1}{1-\mu^{2}}(\varepsilon_{x}-\mu \;\varepsilon_{y}).$$

The maximum normal stress range was 53 MPa. The calculated dynamic lift factor was 1.57. The permissible crane lift coefficient was $\delta_{h}=1.343$; therefore, the calculated dynamic lift coefficient was higher than the permissible lift coefficient.

After adjusting the cross-section of the main beam, measurements were taken after 12 months of operation of the structure. The resulting normal stresses determined by the combination of experiments and calculations did not exceed 110 MPa. Figure 6 shows the normal stress distribution obtained from the measured strain gauge readings after 12 months of operation at the modified cross-section of the main beam at S3 in mode 3. The lift coefficient $\delta_{h}=1.05$ was determined from the results of normal stresses. The value of the lift coefficient in this case was lower than the maximum permissible value, which indicated a significant improvement in the technical condition of main beams. 

After 15 months of operation of the crane since the adjustment of the cross-section of the main beam, another control tensiometric measurement was performed at the same points on the main beam (Figure 7).

The control strain gauge measurement of the main beams after 15 months of operation of the crane since the cross-section adjustment implied that
the resulting normal stress, determined experimentally by calculation, did not exceed 110 MPa;on the basis of experimental measurements, the calculated dynamic lift factor had reached the value of 1.059 and did not exceed the permissible value;the maximum shear stress value obtained from the measured deformation values at S1 with a load of 200,000 kg did not exceed 40 MPa (Figure 8).

However, after checking the structure, local buckling of a part of the welded plates on the main beams was found. For this reason, a further control tensiometric measurement was made seven months after the previous measurement, i.e., 22 months after the rectification of the beam cross-section. The maximum load for this measurement was 111,800 kg in mode 3, because for operational reasons it was not possible to secure a load of 200 t. Stress distributions were therefore transformed by linear extrapolation to the load of 200,000 kg. Figure 9a represents a normal stress distribution at the load of 111,800 kg. Figure 9b represents the transformed time course of normal stress to the weight of 200,000 kg.

The calculated dynamic lift factor did not exceed 1.1. The corresponding value of the shear stress at S1 did not exceed the value of the stress increment of 40 MPa (Figure 10). The character of the normal and shear stresses obtained from the measured values was identical with the character of the stresses measured after 15 months of operation.

Given that the structure was not loaded with a load greater than 130 t, it was concluded that its temporary use in this limited operation mode was possible. 

## 6. Conclusions

In this case, the structure of a bridge casting crane working under conditions of heavy metallurgical operation was the subject of strength and durability analyses. The reason for these analyses was damage to the longitudinal fillet welds between the upper flange and the web of the main beams during the travel of the crane trolley, caused by large wheel pressures. This opened the originally closed main beam profile, which greatly reduced its strength and stiffness. As a result, the lifting capacity and fatigue life of the entire crane had been reduced. These conclusions were confirmed by subsequent computational and experimental analysis. The assessment of fatigue damage cumulation showed the complete fatigue exhaustion of the crane main beam steel structure. Therefore, the rectification of the damaged load-bearing structure was carried out using the appropriate structural modifications. 

Eventually, the above-mentioned modifications resulted in a partial strengthening of the structure with an extension of its service life, but with a time limit of 2 years and a reduced load capacity of 150 t. The prolongation of the service life to more than two years would have required higher costs and more complex design modifications.

The analytical-experimental methods and procedures for assessment of the residual life of load-bearing elements of structures after long-term operation applied in this paper can be used in practice for a number of similar cases. This concerns elements and structures subjected to varying loads that can cause fatigue damage accumulation at critical points and the grow and spread of fatigue cracks leading to fatigue fracture. These elements include in particular load-bearing elements and equipment for bridge, gantry and tower cranes, crane tracks, load-bearing and support structures under machinery and other devices with a dominant rotating effect. The applied methods are also applicable for the fatigue analysis of load-bearing elements with a dominant transverse and longitudinal loading force effect, such as road and railway bridges, etc.

## Figures and Tables

**Figure 1 materials-13-02708-f001:**
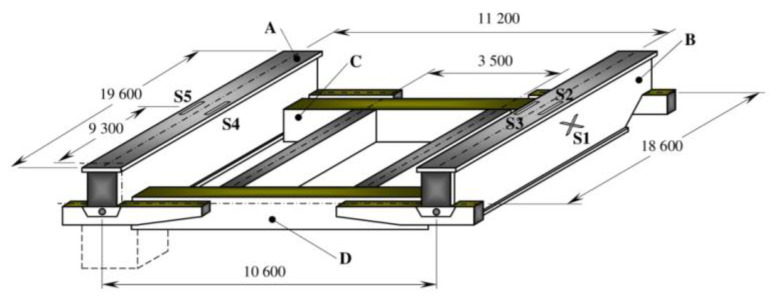
Load-bearing steel structure of the bridge crane with positioning of the strain gauges [mm] (A—left beam, B—right beam, C—rear cross member, D—front cross member).

**Figure 2 materials-13-02708-f002:**
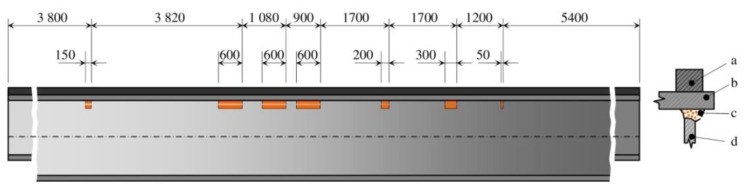
Position and length of fatigue cracks in the longitudinal weld of the main beam [mm] (a—rail, b—flange, c—weld with crack, d—web).

**Figure 3 materials-13-02708-f003:**
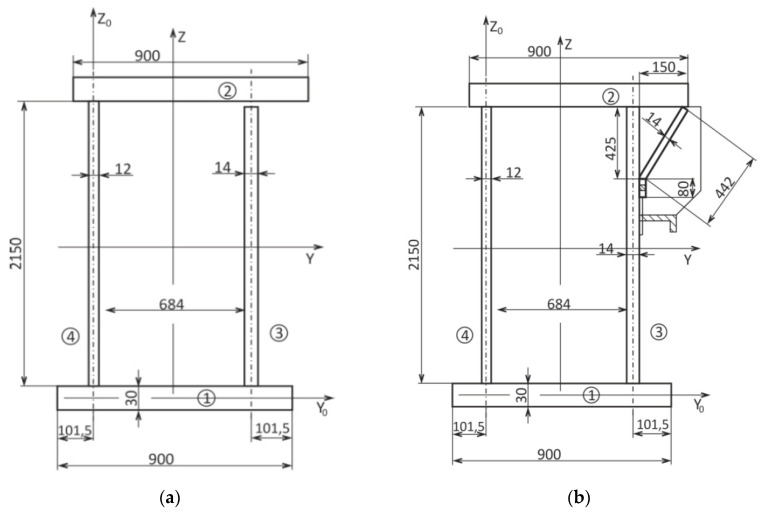
Cross-section of the main beam [mm] (**a**) before treatment, (**b**) after treatment of the cross-section.

**Figure 4 materials-13-02708-f004:**
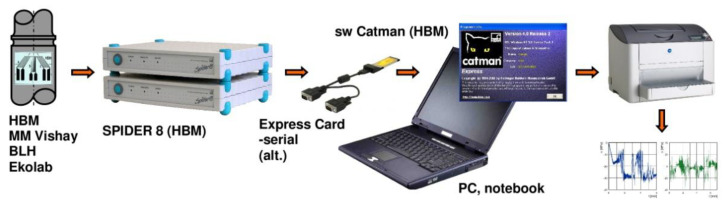
The measuring chain [12].

**Figure 5 materials-13-02708-f005:**
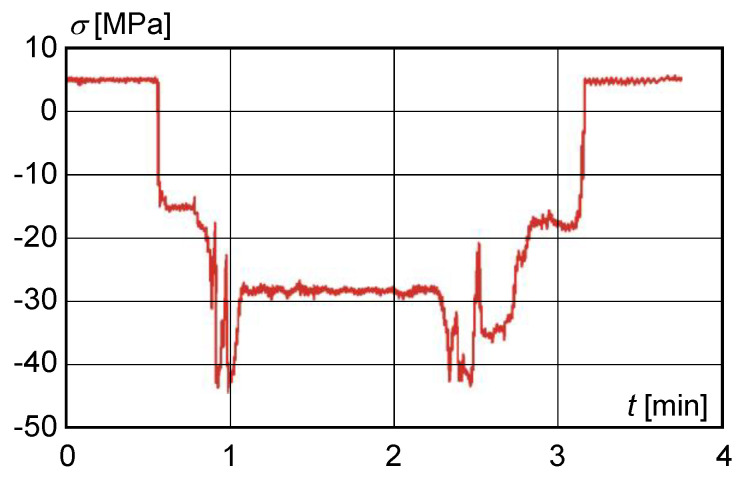
Normal stress distribution obtained from measurements before main beam cross-section adjustment at the S3 strain gauge.

**Figure 6 materials-13-02708-f006:**
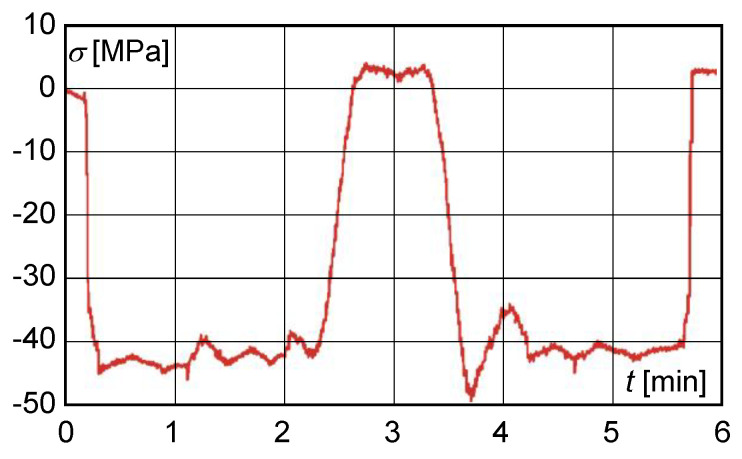
Normal stress distribution obtained from measurements taken after 12 months at the modified cross-section of the main beam at the S3 strain gauge.

**Figure 7 materials-13-02708-f007:**
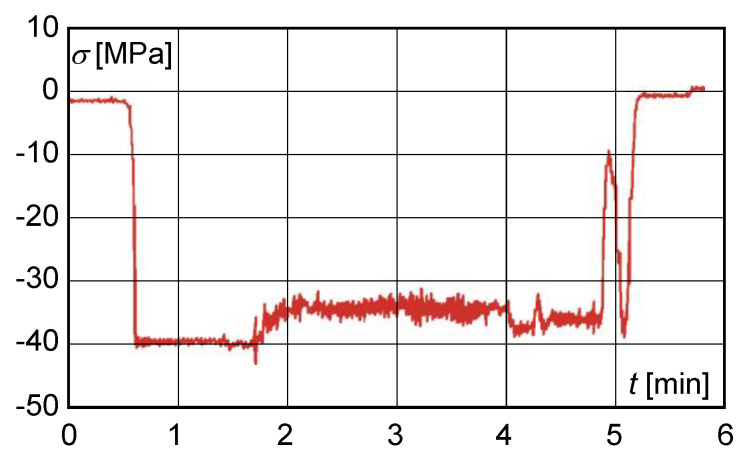
Normal stress distribution obtained from measurements taken after 15 months of operation.

**Figure 8 materials-13-02708-f008:**
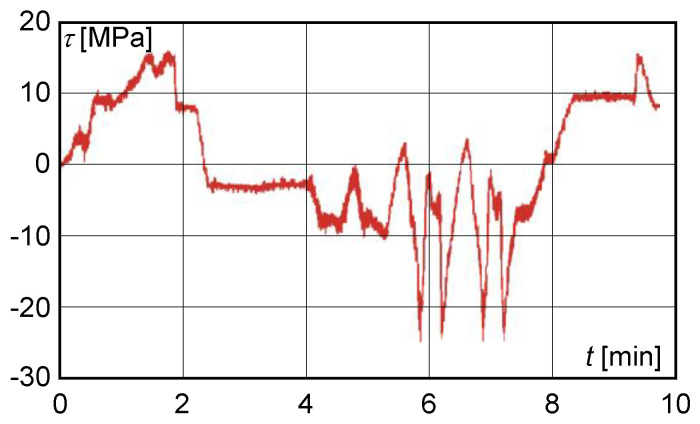
Shear stress behaviour obtained from measurements after 15 months of operation.

**Figure 9 materials-13-02708-f009:**
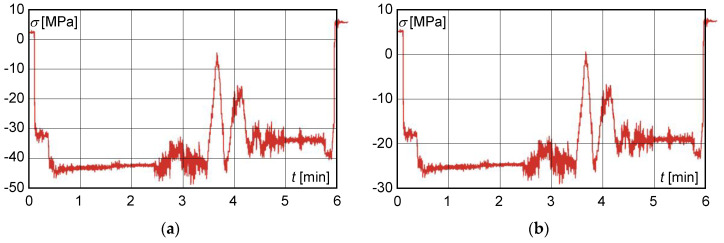
Distribution of normal stress obtained from measurements after 22 months of operation since main beam cross-section modification. (**a**) at the load of 111,800 kg; (**b**) at the transformed to the weight of 200,000 kg.

**Figure 10 materials-13-02708-f010:**
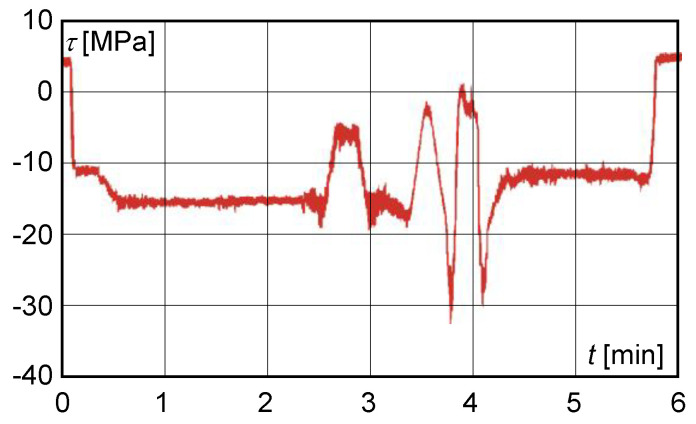
Transformed shear stress time course at S1 after 22 months of operation.

**Table 1 materials-13-02708-t001:** Geometric characteristics of the main lift beam before and after cross-section adjustment.

Description	Symbol and Unit	Before Modification of the Cross-Section	After Modification of the Cross-Section
Cross-section surface	$A\;\;\left({\mathrm{mm}}^{2}\right)$	109.90 × 10^3^	117.70 × 10^3^
Cross-section module at the bend towards the y-axis	$W_{y}\;\;\left({\mathrm{mm}}^{3}\right)$	77.56 × 10^6^	86.14 × 10^6^
Cross-section module at the bend towards the z-axis	$W_{z}\;\;\left({\mathrm{mm}}^{3}\right)$	21.82 × 10^6^	24.88 × 10^6^
Area enclosed by the centre line	$A_{S}\;\;\left({\mathrm{mm}}^{2}\right)$	1519.46 × 10^3^	1556.59 × 10^3^
Main quadratic moment	$J_{y}\;\;\left({\mathrm{mm}}^{4}\right)$	85,700.00 × 10^6^	90,680.00 × 10^6^
	$J_{z}\;\;\left({\mathrm{mm}}^{4}\right)$	10,430.00 × 10^6^	11,360.00 × 10^6^

**Table 2 materials-13-02708-t002:** Partial factors taking into account the uncertainties of load estimation.

Load Coefficient		Dynamic Coefficient	
From its own weight	$\gamma_{q}=1.1$	Travelling	$\delta_{t}=1.1$
From the load	$\gamma_{lo}=1.2$	Lifting	$\delta_{h}=1.3+0.39v_{h}$
From forces of inertia	$\gamma_{i}=1.1$	For the main lift (200 t)	$\delta_{h}=1.343$
From the path crossing	$\gamma_{tp}=1.0$	For the auxiliary lift (50 t)	$\delta_{h}=1.352$

**Table 3 materials-13-02708-t003:** Calculated force values and stresses on the main beam after cross-section adjustment.

Heading	$T_{z}$ (kN)	$M_{y}$ (kNm)	$M_{z}$ (kNm)	τ (MPa)	σ (MPa)	$\sigma_{red}$ (MPa)
In the centre of the beam span without crossing	1184.00	10,357.80	245.16	14.10	130.10	132.40
At the point of joining the beam to the crossbeam without crossing	2261.80	6141.00	205.40	26.90	79.54	92.2
In the centre of the beam span with crossing	963.50	8571.80	196.20	11.50	107.40	109.2
At the point of joining the beam to the crossbeam with crossing	1930.70	5034.70	499.20	22.90	78.50	88

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
