# Peer review of "Experimental Assessment of Time-Limited Operation and Rectification of a Bridge Crane"

_materials, 2020, doi:10.3390/ma13122708_

Round 1

Reviewer 1 Report

Please see attached

Author Response

Dear Reviewer,

Thank you for your time, very helpful remarks and suggestions that really helped us to improve the paper and to make it more attractive for readers, especially for the international audience. We have made substantial changes to the manuscript, taking into account all comments.  Please find below our response to each of the comments.

Note, changed fragments in the article body are marked in green.

Reviewer 2 Report

Submitted manuscript entitled “Assessment of the time limited service of beam of lifting structure" shows capability of load-bearing elements and structures for long-term operation and predicting their total or residual service life. Topic is interesting and I recommend it to publication. I think paper is written in good enough language, but maybe English language verification should be done by the right person for English spelling and grammar. Technique, technology and research methods used in the work are adequate. Methods and obtained results prove founded thesis and show originality of the manuscript.

Author Response

(The authors gave the same response as above.)

Reviewer 3 Report

The manuscript entitled "Assessment of the time limited service of beam of lifting structure” provided a case study to describe the rectification process of the damaged bridge crane working in heavy metallurgical operation. The manuscript discussed an interesting topic about retrofitting a damaged existing structure to extend its fatigue life. Therefore, this reviewer recommends accepting it after considering the following comments.

Technical comments:

  • The manuscript could benefit greatly from professional editing to improve technical writing and English. For example, the title of the manuscript should be improved Grammarly.
  • Lines 20-29: Un-necessary introduction that should not be included in the abstract. I mean these sentences should be concise.
  • Some of the main conclusions should be highlighted in the abstract.
  • Equation 1: The dominator of the third term in equation 1 does not make sense for me. I believe that the dominator should be Rx time Ry.
  • Line 153: This sentence should be improved Grammarly.
  • Line 163: How did you determine this number of cycles. The authors should provide information about this point.
  • Line 184: Longitudinal view of the beam to show the length of the bent sheet is recommended. Did the authors provide this bent sheet at locations of the fatigue cracks where the cross-section is open? or through the full length of the beam?
  • Lines 191-192: The authors should be more specific at this point. Do you mean that the main beam is a built-up section made from steel plates (EURO S235JRG2) with a thickness of 25 mm?
  • Line 192: What do you mean by "computational strength"? Do you mean yield or ultimate strength?
  • Lines 193-194: This sentence should be rephrased to list all kinds of loads which taken into consideration.
  • Line 201: Do you mean that this force will cause shear force and torsion?
  • Equation 2: The authors should cite a reference to support this equation.
  • Line 220: What do you mean by "relative deformations"? Typically, strain gages are used to measure strains.
  • Lines 224-225: Why these levels of loading? The authors should relate the loading level to the beam capacity. This will help to sense the measurements.
  • Line 234: Are they sensors or strain gages?
  • Lines 241-244: This reviewer highly recommends providing the formula from the theory of elasticity that used to calculate the normal stress.
  • Line 242: What do you mean by "flexible deformations"? Do you mean elastic deformations?
  • Line 266: How did the authors obtain shear stress from the measured deformation or strains? I believe that shear stress cannot be obtained with only one strain gage in one-direction. So, how did the authors get this value from the measured results?
  • Lines 269-270: This sentence is not clear to me. I cannot understand what you mean.
  • Lines 280-281: How did the authors measure shear stress?
  • Line 309: What do you mean by this sentence?
  • Lines 310-317: Why the authors mention two sections for the Acknowledgements?

Author Response

(The authors gave the same response as above.)

Reviewer 4 Report

Cross-section of the main beam after treatment could be explained. Why use this new dimensions. Explain the new calculation of new dimensions.

Author Response

(The authors gave the same response as above.)
